# CoLa-DCE – Concept-guided Latent Diffusion Counterfactual Explanations

## Abstract

Recent advancements in generative AI have introduced novel prospects and practical implementations. Especially diffusion models show their strength in generating diverse and, at the same time, realistic features, positioning them well for generating counterfactual explanations for computer vision models. Answering "what if" questions of what needs to change to make an image classifier change its prediction, counterfactual explanations align well with human understanding and consequently help in making model behavior more comprehensible. Current methods succeed in generating authentic counterfactuals, but lack transparency as feature changes are not directly perceivable. To address this limitation, we introduce Concept-guided Latent Diffusion Counterfactual Explanations (CoLa-DCE). CoLa-DCE generates concept-guided counterfactuals for any classifier with a high degree of control regarding concept selection and spatial conditioning. The counterfactuals comprise an increased granularity through minimal feature changes. The reference feature visualization ensures better comprehensibility, while the feature localization provides increased transparency of "where" changed "what". We demonstrate the advantages of our approach in minimality and comprehensibility extensively across multiple datasets, classification models, and diffusion models and provide insights into how our CoLa-DCE explanations help comprehend model errors like misclassification cases.

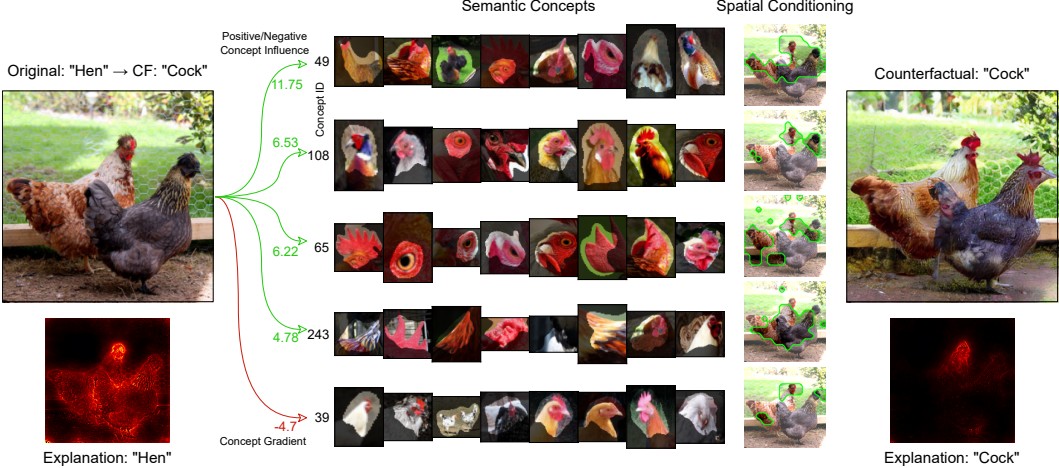

Figure 1: Example image of a concept-based counterfactual with CoLa-DCE consisting of a selection of concepts with reference samples, a localization map per concept indicating the concept regions, and the generated counterfactual.

## 1 Introduction

Recent advancements in generative models have sparked new interest in counterfactual explanations for computer vision tasks Augustin et al. (2022); Jeanneret et al. (2023a); Farid et al. (2023). By

answering what would need to change to induce a different outcome, counterfactual explanations are motivated by research in psychology and the social sciences highlighting the alignment of counterfactuals to human reasoning Lewis (1973); Byrne (2007). While current development efforts in eXplainable Artificial Intelligence (xAI) often focus on technical feasibility rather than on the alignment with human understanding of a Deep Neural Network (DNN) model, counterfactuals provide an opportunity for the user to contemplate alternative model outputs Miller et al. (2017). In the image domain, a human inspector can directly compare an original image with its counterfactual to derive which differences induce a prediction change in the model under test. Key requirements for the counterfactual to be deemed a plausible alternative are the consistency with the user's beliefs, as being realistic, and a minimal effort for changing towards the counterfactual Byrne (2007). The minimality constraint expresses a more likely transition due to a smaller image alteration, while additionally, the decision boundary between both classes can be better estimated.

Specifically for image manipulations, diffusion models have demonstrated their strengths in generating realistic high-resolution images with diverse features within the data distribution Ho et al. (2020); Dhariwal & Nichol (2021); Rombach et al. (2022); Ho & Salimans (2022). Thus, they serve as an ideal tool for generating counterfactual images Augustin et al. (2022); Farid et al. (2023); Jeanneret et al. (2023a). While previous works on diffusion-based image counterfactuals find optimizations regarding all features in an image or a local area inside an image, it is often unclear which features precisely change toward the counterfactual and how they relate to the model prediction. Especially with many slight feature changes in an image, tracking the changes and comprehending the decision boundary based on these features becomes unfeasible. Considering the example of the "hen" in Figure 1, every part of the animal, e.g., head, feathers, and color, as well as background features like the flooring, could yield significant changes towards the counterfactual class without being recognizable to the user.

In comparison, humans tend to perceive the minimal differences in counterfactuals rather in semantic than in pixel space and prefer representative differences Delaney et al. (2023). This motivates the yet missing strategy of defining minimality semantically in the number of semantic features changed and further encourages a concept-based approach enforcing understandable semantic alterations.

With our Concept-guided Latent Diffusion Counterfactual Explanations (CoLa-DCE), we solve both problems: We guide the counterfactual generation with a restricted number of semantic concepts, further enabling a high level of control by concept selection. We additionally include feature visualization capabilities, allowing for direct comprehensibility of features that represent the difference between the original and the counterfactual class. Hereby, CoLa-DCE provides semantic as well as spatial guidance and visualization, simultaneously enabling control and better transparency. Our contributions are:

1. We introduce CoLa-DCE for the diffusion-based generation of counterfactuals using a semantic concept-guidance. We show how local counterfactual targets and concept-guided feature changes derived from the classifier's perception increase the quality of the counterfactuals.

2. We extend our concept guidance with spatial conditioning and reveal the semantic and localized image changes with transferring methods for concept visualization and concept localization maps, resulting in more transparent and more comprehensible counterfactuals highlighting the image changes.

3. We show how our CoLa-DCE samples help in model debugging by making cases of misclassification more understandable. The semantic concept visualization provides strategies for feature-based model and/or dataset adaptations.

The source code will be published at `github.com/anonimous-url/cola-dce`.

## 2 BACKGROUND

Diffusion models Ho et al. (2020) evolve from the idea of gradually adding small amounts of Gaussian noise to an image in a forward process, which can then be gradually reversed by learning the respective backward process. Given scalar noise scales $\alpha_t{}_{t=1}^T$ with $T$ denoting the number of time steps and an input image $x_0$, the noisy image representations $x_t$ for the forward diffusion process

can be computed with:

$$x_t = \sqrt{\alpha_t}x_0 + \sqrt{1 - \alpha_t}\epsilon_t, \quad \text{where} \quad \epsilon_t \in \mathcal{N}(0, \mathbf{I}). \tag{1}$$

Based on the current noise sample $x_t$ and time step $t$, a modified U-Net Ronneberger et al. (2015) can be used for estimating the noise $\hat{\epsilon}_t$, which was added at that time step:

$$\epsilon_\theta(x_t, t) \approx \hat{\epsilon}_t = \frac{x_t - \sqrt{\alpha_t}x_0}{\sqrt{1 - \alpha_t}}. \tag{2}$$

The original image $x_0$ can be approximately predicted, when rewriting Equation 2 as:

$$\hat{x}_0 \approx \frac{x_t - \sqrt{1 - \alpha_t}\epsilon_\theta(x_t, t)}{\sqrt{\alpha_t}}. \tag{3}$$

Sampling methods like the DDIM sampling Song et al. (2021) speed up the image generation by estimating multiple timesteps and can be used to sample the next less-noisy representation $x_{t-1}$:

$$x_{t-1} = \sqrt{\alpha_{t-1}}\frac{x_t - \sqrt{1 - \alpha_t}\hat{\epsilon}_t}{\sqrt{\alpha_t}} + \sqrt{1 - \alpha_{t-1} - \sigma_t^2}\hat{\epsilon}_t + \sigma_t\epsilon_t. \tag{4}$$

Latent diffusion models Rombach et al. (2022) decrease the dimensionality of the input by incorporating an additional encoder-decoder architecture. The encoder derives a dense representation of the data point so that the diffusion process can be applied in the dense feature space. The generated output is decoded into an observable image afterward.

For guiding the image generation with an external classifier, Dhariwal & Nichol (2021) introduces classifier guidance with a scaling factor influencing the trade-off between the accuracy and diversity of generated images. Classifier-free diffusion guidance Ho & Salimans (2022) separates the conditioning into an unconditional part and a conditional part, where the difference between both parts can be used as an implicit classifier score:

$$\nabla_x \log p_\eta(x|c) = \nabla_x \log p(x) + \eta \nabla_x \log p(c|x). \tag{5}$$

This gradient-based scoring for guiding the diffusion process by both external and implicit classifiers can be further utilized to constitute the counterfactual generation by actively shaping the gradient.

## 3 RELATED WORK

### 3.1 COUNTERFACTUAL IMAGE GENERATION

Generating counterfactuals in the image domain requires the capability to modify existing or generate new features in an image. While approaches like Filandrianos et al. (2022) compare images by the set of assigned attributes and define the counterfactual to be the near miss from the used reference dataset, most approaches directly modify the base image itself. Counterfactual Visual Explanations (CVE) Goyal et al. (2019) replaces feature regions in an image with matching image patches from a distractor image of the counterfactual class. Other works directly optimize an input image by minimizing a loss, shifting the classification towards the counterfactual class while keeping the image changes minimal Santurkar et al. (2019); Augustin et al. (2020). SVCE Boreiko et al. (2022) yields further improvements to the optimization by combining the L1- and L2-norm to acquire a balance between non-sparse and too-sparse feature changes. However, directly optimizing the image requires a robust classification model. Another group of methods is based on autoencoder architectures to control the optimization in a disentangled latent space Rodríguez et al. (2021) or to apply modifications in a simplified interpretable space Zemni et al. (2023). Yet, with diffusion models, better possibilities for high-quality feature generation exist.

DiME Jeanneret et al. (2023a) introduces diffusion models for generating counterfactuals, where the classification model guides the diffusion process. However, DiME is limited to robust classifiers explicitly trained on noisy images. ACE Jeanneret et al. (2023b) is a two-step process consisting of computing pre-explanations and refining them. A localization mask for the most probable feature change is computed before repainting the image by combining the generated counterfactual within the mask with the original image outside.

Diffusion Visual Counterfactual Explanations (DVCE) Augustin et al. (2022) relaxes the constraint for the classifier to be robust by including an additional adversarially robust classifier. Aligning the gradients of both models with a cone projection robustifies the diffusion guidance. However, generated features might be induced by the robust classifier rather than the original classifier, decreasing the validity in explaining the original classifier. Latent Diffusion Counterfactual Explanations (LDCE) Farid et al. (2023) overcomes the requirement of having a robust classifier by constructing a consensus mechanism for aligning the gradient of the external classifier with the gradient of the implicit classifier of the diffusion model directly. However, feature changes are hard to track due to the optimization on all features.

Although the previous works are able to generate realistic counterfactual images, the resulting counterfactuals lack transparency regarding which features have been changed and how the change is reflected in the parameters of the target model. To our knowledge, the image domain has not considered a concept-based approach that guides feature changes on a semantic concept level and enforces minimality by restricting the number of feature changes. Concept-based counterfactuals yield the opportunity to improve transparency and comprehensibility for the user while being semantically more similar to the original image.

## 3.2 LOCAL CONCEPT ATTRIBUTION

Layer-wise Relevance Propagation (LRP) Bach et al. (2015) describes a local attribution method that backpropagates a modified gradient to assign pixel-wise importance scores for an input based on a selected target class. Concept-wise Relevance Propagation (CRP) Achtibat et al. (2023) extends LRP to the concept space by defining the encoding of every single neuron or channel in the latent space as a concept. During the attribution backward pass, a concept mask is applied, which filters the attribution for a single channel so that only the attribution for the selected channel is retained. When inspecting the channel-constrained explanations for multiple samples, denoted as Relevance Maximization Achtibat et al. (2023), a semantic meaning describing a concept can be assigned to the channel. Our approach utilizes the generalization of the latent space masking for a gradient manipulation and applies Relevance Maximization to visualize the determining concepts.

## 4 CONCEPT-GUIDED LATENT DIFFUSION COUNTERFACTUAL EXPLANATIONS

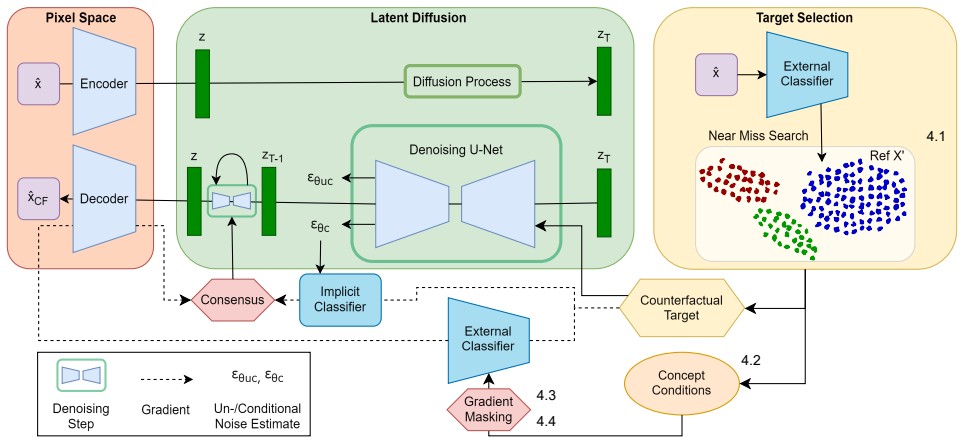

Figure 2: A simplified overview of the model architecture for our CoLa-DCE approach, including the target selection (right) and the concept-conditioning for guiding the diffusion denoising (middle).

Our CoLa-DCE method consists of three main improvements to current diffusion-based image counterfactual methods. In step 1, local sample-based targets are derived based on the model perception. Step 2 consists of concept conditioning to guide the image adaptation using a selection of concepts, while step 3 adds spatial conditioning to the selected concepts. Thus, concept and spatial conditioning selectively modify the classifier gradient before conditioning the diffusion generation process.

### 4.1 LOCAL COUNTERFACTUAL TARGETS

To select the counterfactual target class, we use the model's perception of the respective data sample and compare it to the perception of a reference dataset $X'$. The model perception can hereby be derived by either computing the activation of the model for each sample in a selected layer or by computing the intermediate attribution using a local xAI method like LRP Bach et al. (2015). As the model perception of the data shall be represented, the class predictions of the model are used to determine class affiliation.

$$y_c = f(argmin_{x' \in X'} d(\kappa(x'), \kappa(\hat{x})) \quad \text{and} \quad f(x') \neq f(\hat{x})) \tag{6}$$

For a new sample $\hat{x} \in \hat{X}$ that we want to generate a counterfactual for, we derive the model prediction and feature space encoding $\kappa(\hat{x})$ and compare it to the encodings of the reference dataset. Hereby, based on the feature space encodings, the closest reference point with a differing class prediction is extracted, resembling the near miss approach Rabold et al. (2022). The counterfactual target $y_c$ is then defined as the predicted class of the reference point $x'$.

---

**Algorithm 1** CoLa-DCE algorithm for sample $x_i$ with $k$ concepts and class condition $c$

---

$\hat{y} \leftarrow NearMiss(x_i)$          # Compute counterfactual target
$grad \leftarrow \nabla_x p(x|\hat{y})$          # Compute gradient to counterfactual
$\lambda_0 ... \lambda_k \leftarrow topk(grad, k)$          # Extract k most-important concepts
$\theta_0 ... \theta_k \leftarrow get\_masks(\lambda_0, ..., \lambda_k)$      # Compute concept (and spatial) constraints
**for** t=T,...,0 **do**
    $cls\_score \leftarrow \sqrt{1 - \alpha_t} \nabla_{z_t} L(f(\hat{x}_0|\hat{y}, \theta_1...\theta_k), c)$     # Apply LDCE with constraints
    $z_{t-1} \leftarrow ApplyLDCE(cls\_score)$
**end for**
$x_i^{CF} \leftarrow \mathcal{D}(z_0)$          # Decode reconstruction

---

### 4.2 CONCEPT SELECTION

For a selected counterfactual target class $y$, the gradient $\nabla_x p(x|y)$ of a sample $x$ can be extracted in each network layer. For the selected layer $l$, the intermediate gradient is summed over the spatial dimensions to obtain a one-dimensional representation over the channels, which are expected to encode a particular concept each Achtibat et al. (2023). Taking the absolute value of the summed gradients, the top-$k$ concepts with $k \in \mathcal{N}(1, K)$, and $K$ denoting the overall number of channels, are selected, which are per gradient most likely to induce a change towards the counterfactual class. The concepts can be visualized using a feature visualization method like CRP Achtibat et al. (2023).

### 4.3 CONCEPT CONDITIONING

Based on the LDCE Farid et al. (2023) algorithm, we apply an additional concept conditioning functionality concerning the selected concepts. The conditions require precomputation and remain fixed during the counterfactual generation, as adapting the conditions to each single generation step leads to changing concepts in each step.

Instead of using the complete gradient of the external classifier $\nabla_x p(x|y)$ for target $y$, the conditioned gradient with regards to the selected concepts $\lambda_1, ..., \lambda_k$ with binary constraints $\theta_1, ..., \theta_k$ is computed. With the selected layer $l$ splitting the model into two parts $p(x|y) = h(g(x|y)|y)$, the conditioned gradient is computed as:

$$\nabla_x p(x|y, \theta_1...\theta_k) = \nabla_x(h(g(x))|y, \theta_1...\theta_k)$$
$$= \delta(\nabla_{g(x)} h, \theta_1...\theta_k) \cdot \nabla_x g$$
$$\text{with} \quad \delta(\nabla_{g(x)} h, \theta_1...\theta_k)_j = \begin{cases} \nabla_{g(x)} h_j, & \text{if } j \in \{\theta_1, ..., \theta_k\} \\ 0, & \text{otherwise} \end{cases} \tag{7}$$

with $\delta$ indicating binary masking the latent space gradient in the selected layer. The masked latent gradient can be backpropagated to the input without further constraints.

## 4.4 Spatial Conditioning

While the introduced concept conditioning focuses on semantic features that need to change, the spatial dimensions in the feature layer of choice state where the selected features are most likely to change. We assume that each feature should be only changed at a single location or that the gradient towards these features is approximately equal in equivalent locations. Therefore, we add binary masking to the spatial dimensions similar to Equation 7 based on the gradient for the selected features, which sets gradients below a threshold $\eta$ to zero. The binary mask can additionally be upscaled to the input scale like in Net2Vec Fong & Vedaldi (2018), yielding additional information about where a specific concept is expected to change towards the counterfactual. The spatial conditioning minimizes the feature change by restricting it locally while contributing to comprehensibility by providing feature localization.

## 5 Results

We test our approach on the ImageNet Deng et al. (2009) validation dataset using multiple pre-trained models provided by Torchvision: a VGG16 Simonyan & Zisserman (2015) with and without batch normalization, a ResNet18 He et al. (2016), and a ViT model Dosovitskiy et al. (2021). For deriving appropriate targets, 90% of the validation data is used as a reference dataset, while counterfactuals for the evaluation are generated on the remaining 1000 samples, including all ImageNet classes. We inherit the parametrization parameters from LDCE Farid et al. (2023). Showing the applicability to different datasets, additional counterfactuals for Oxford Pets Parkhi et al. (2012) and Flowers Nilsback & Zisserman (2008) can be found in Appendix A.3.

As there exists no ground truth for counterfactual examples, a rough estimate regarding the quality can only be assessed via quantifying desired properties as the minimality and the accuracy. We align our evaluation with Farid et al. (2023) and compute the FID score Heusel et al. (2017) as well as the L1 and L2 norm between the original and counterfactual image to measure their semantic and pixel-based distance, denoting the minimality. The flip ratio (FR) determines the accuracy by measuring how often the classifier predicts the counterfactual class for the generated sample.

As an additional optimization measure, we suspend the concept conditioning for the last 50 generation steps of the diffusion process. While coarse semantic features are expected to be generated within the first steps of the diffusion process, the last steps incorporate an image refinement, e.g., by completing and connecting edges. When suspending the conditioning towards the end of the generation, visible semantic changes are not perceivable, but the image is classified more accurately. This can also be seen in a consistent FID score and an improved flip ratio.

### 5.1 Selecting a local target results in improved counterfactuals

While LDCE Farid et al. (2023) uses WordNet Miller (1995) to derive counterfactual targets based on the semantic similarity between labels, we suggest using the classifier's perception of the local input. Selecting a target layer, the classifier-internal representation of a data point can be extracted via the activation or the attribution using a local xAI method. Based on the encodings of a reference dataset, the sample with minimal distance and differing class prediction to the encoded target sample is extracted. It's prediction is chosen as counterfactual target. The approach is related to the concept of near misses Rabold et al. (2022).

Table 1 shows the influence of the target selection on the generated samples' quantitative performance metrics. Choosing a local (sample-based) counterfactual target on a near-miss basis leads to an improved flip ratio and confidence in all settings, demonstrating a nearer decision boundary and more superficial change between the original and target class. However, retrieving the target via the activation may lead to a slightly increased FID compared to the baseline, as some counterfactual targets have no semantic connection to the original class. Thus, a more substantial semantic change is required. Using the intermediate LRP Bach et al. (2015) attribution yields substantial improvements in the minimal change needed while simultaneously achieving high flip ratios. This indicates semantically similar counterfactuals close to the original images. Including the model's classification in the intermediate attribution rather than only considering the activation up to the selected layer may better represent how the features in the layer are connected toward the output, comprising top-level semantics between classes. Thus, fewer feature changes are necessary. Including the results

Table 1: Quantitative comparison showing the effect of the target selection on the generated counterfactuals using the LDCE method in comparison to our CoLa-DCE method ($k$=20).

| Model Setting | | | | FID ↓ | L1 ↓ | Flip Ratio ↑ | Confidence ↑ |
|---|---|---|---|---|---|---|---|
| Model | Method | Target | Layer | | | | |
| VGG16bn | LDCE | Base | - | 55.46 | 12458 | 0.851 | 0.81 |
| VGG16bn | LDCE | Act | feat.37 | 59.12 | 12456 | 0.936 | 0.89 |
| VGG16bn | LDCE | Attr | feat.37 | 45.56 | **12443** | **0.956** | **0.92** |
| VGG16bn | CoLa-DCE | Attr | feat.37 | **44.43** | 13915 | 0.821 | 0.81 |
| ResNet18 | LDCE | Base | - | 55.86 | 12518 | 0.846 | 0.79 |
| ResNet18 | LDCE | Act | 4.1.c1 | 57.46 | 12502 | **0.96** | **0.91** |
| ResNet18 | LDCE | Attr | 4.1.c1 | 46.28 | **12465** | 0.957 | **0.91** |
| ResNet18 | CoLa-DCE | Attr | 4.1.c1 | **44.86** | 13933 | 0.846 | 0.84 |
| ViT | LDCE | Base | - | 59.48 | **12533** | 0.833 | 0.81 |
| ViT | LDCE | Act | encoder | 53.75 | 14024 | 0.913 | 0.88 |
| ViT | LDCE | Attr | encoder | 53.24 | 14028 | **0.917** | **0.89** |
| ViT | CoLa-DCE | Attr | encoder | **53.21** | 14003 | 0.847 | 0.83 |

of our CoLa-DCE method, even closer counterfactuals are generated with flip ratios on par with the LDCE baseline. Reconsidering the hard constraint on the number of concepts, damping the gradient signal, CoLa-DCE yields much more transparent counterfactuals while still being competitive to the baseline.

## 5.2 THE NUMBER OF CONCEPTS IS A TRADEOFF BETWEEN ACCURACY AND COMPREHENSIBILITY

Since a counterfactual explanation should depict the minimal semantic change in an image that causes a classifier to change its prediction, we assume that the minimal semantic change can be expressed by the number of changed features or concepts. While generally concept-based approaches in xAI mostly use a handful of concepts for best comprehensibility Zhang et al. (2021); Achtibat et al. (2023); Dreyer et al. (2023); Kim et al. (2023), restricting the latent space gradient in our case from multiple hundred to very few channels significantly reduces the gradient for guiding the diffusion process. We perform a quantitative study to assess how the number of concepts influences the performance in obtaining reliable results regarding accuracy and minimality.

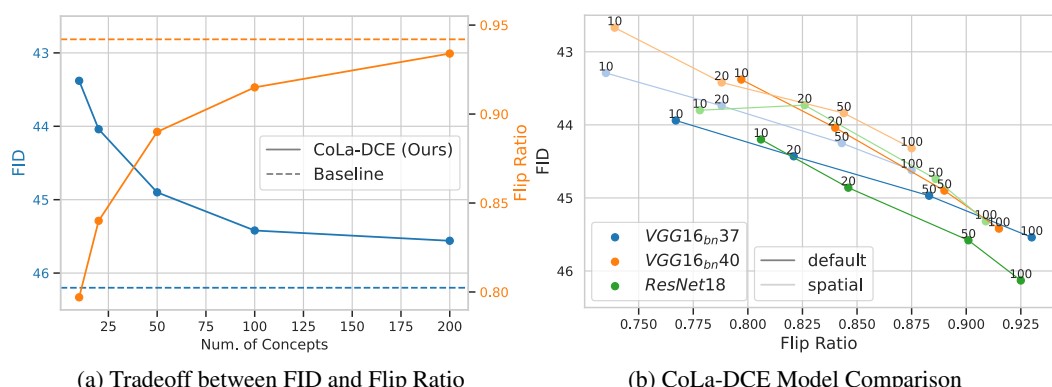

(a) Tradeoff between FID and Flip Ratio  (b) CoLa-DCE Model Comparison

Figure 3: Quantitative evaluation for specifying the tradeoff between the number of concepts and the quantitative measures as flip ratio and FID. The results in 3a are derived for the VGG16bn with target layer `feat.40`.

Figure 3 depicts the relationship between the number of concepts, the FID similarity, and the flip ratio. Restricting the number of concepts leads to an improved FID (minor change) while the flip ratio decreases. The restriction of the gradient causes the image to change in fewer features, but

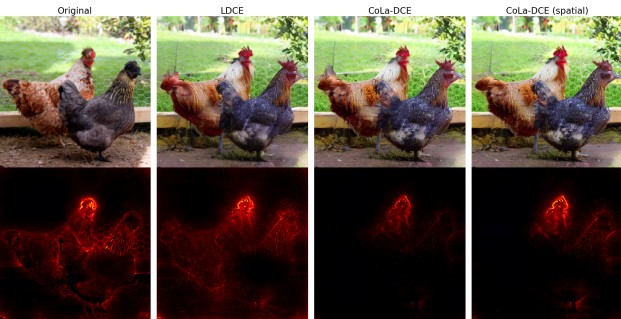

Figure 4: CoLa-DCE explanations ("water ouzel" to "red-backed sandpiper") with a differing number of concepts $k$ and and the VGG16bn with concept layer 40. Limiting the concept number induces more fine-grained feature perturbations than the baseline LDCE, flipping the shown bird completely.

the force pushing the sample towards the counterfactual class is also attenuated. However, a good performance $> 75\%$ regarding the flip ratio can already be achieved with only ten concepts, while the FID score outperforms the baseline. Thus, CoLa-DCE offers concept-based transparency and control without losing much detail or accuracy. Figure 3b depicts the tradeoff between minimality and accuracy for multiple model architectures and settings. Adding spatial constraints per concept results in slightly degraded flip ratios, compensated by an improved FID. Figure 4 shows an example of how the number of concepts influences the counterfactual generation. Restricting the concepts leads to minor changes that alter the target object semantically. In contrast, multiple hundred concepts and the LDCE baseline induce an alteration of the image composition by, e.g., generating new objects like the vertically flipped bird evolving from the upper part of the original bird.

## 5.3 Spatial constraints per concept improve the focus

Figure 5: Comparison of the counterfactual images and their explanations for LDCE and our proposed method CoLa-DCE w/o and with spatial constraints.

Assuming each feature is locally restricted and may only be modified in the most probable region(s), we add spatial constraints per concept by thresholding the gradient. Considering the example of Figure 1, image modifications towards the cockscomb are only reasonable near the head of the hen so that the concept-based gradient can be set to zero in all other regions. Figure 5 shows the difference in the generated counterfactuals for the spatial conditioning and basic CoLa-DCE compared to the LDCE baseline. Compared to LDCE, CoLa-DCE yields much more sparse explanations, highlighting fewer and more concentrated feature changes in the image. With added spatial constraints, a stronger focus in the explanation becomes apparent, either having more sparse explanations or reflecting a stronger focus on single semantic features. Performance-wise, the spatial conditioning further decreases the FID for the better, while only slight drawbacks regarding the flip ratio occur.

## 5.4 How can concept-based counterfactuals help in explaining model failures?

Counterfactuals are especially useful when explaining samples at the classifier's decision boundary between two classes. When misclassified samples and their correctly classified counterfactuals are

inspected using our CoLa-DCE approach, the root cause of the misclassification in terms of identified or missing features becomes apparent. Figure 6 describes a misclassification case where the original image lacks specific evidence of belonging to the label "brambling". The sample seems to represent a rare case of the class where the classifier is missing essential concepts shown in the CoLa-DCE explanation for a correct classification. Hence, a dataset or model adaptation is required.

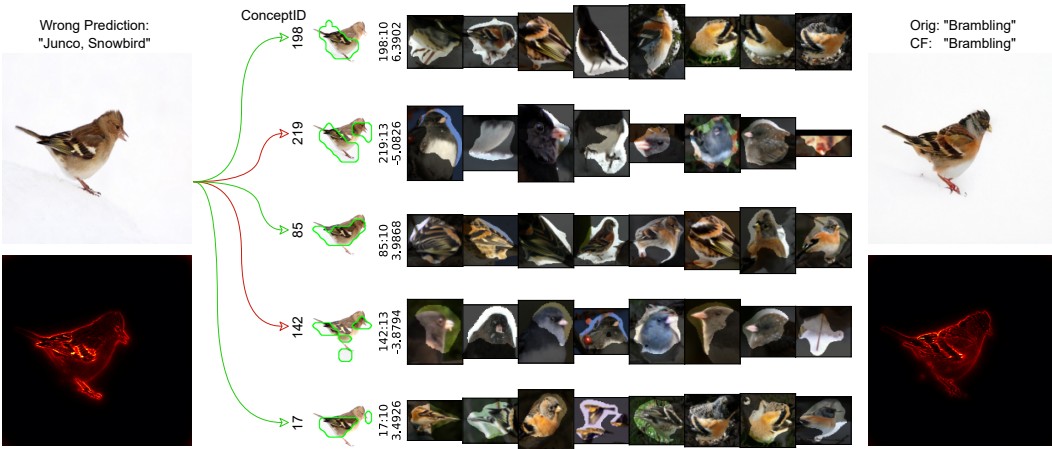

Figure 6: A CoLa-DCE explanation for a misclassified sample, which the VGG16bn classifies as "Junco, Snowbird". To classify the input correctly as "Brambling", the orange chest color, a slightly different feather pattern, and a gray-blueish head color are missing. Besides, the head and beacon shall look less similar to the class "Junco, Snowbird".

## 5.5 Validity: Do the concepts align with the image modifications towards the counterfactual?

Testing the validity of our approach considering the selected concepts, we review whether the change from the original to the counterfactual image targets the selected concepts. The difference in the intermediate attributions of both original and counterfactual images signifies the difference in the importance of the concepts for the respective predictions. We assume the channels with the highest difference to align with the $k$ selected concepts. For estimating the relative alignment, we compute the ratio of the difference $|attr_{counterfactual} - attr_{original}|$ for the selected concepts to the top-$k$ values. The same ratio with $k$ randomly selected concepts is computed for comparison. The results in Figure 7 clearly validate the concept-based approach, as the meaningful change towards the counterfactual can evidently be assigned to the selected concepts for both the VGG16bn and the ResNet18. Due to the redundancy of similar feature encodings in computer vision models, a change in one feature is expected to influence multiple channels in the latent space. Thus, it is reasonable that the selected features do not perfectly align with the top-$k$ concepts with the highest attribution difference.

## 6 Limitations

As ground truth information of an optimal counterfactual image does not exist, only heuristics containing desired properties can be optimized. However, the right balance between minimally deviating the image while maximizing the flip ratio depends on a rough estimate of the user's preferences. Like in LDCE, the influence of the external gradient and the reconstruction accuracy need to be fine-tuned. Including the influence of the diffusion model, we acknowledge that the diffusion's ability to accurately reconstruct an image and generate similar concept information as the external classifier highly influences the counterfactual quality.

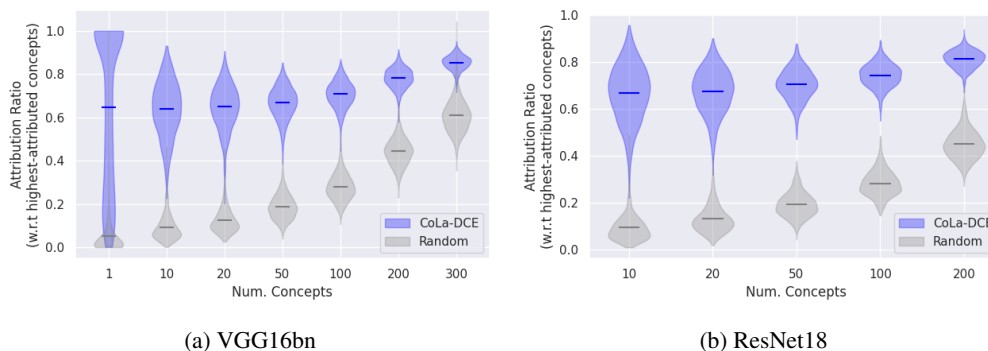

(a) VGG16bn        (b) ResNet18

Figure 7: The validity evaluation computes the ratio of attribution difference between counterfactual and original image for the selected concepts concerning the concepts with the strongest attribution difference. A 1.0 ratio describes the optimal fit of selected concepts. Our CoLa-DCE method shows a strong connection between selected concepts and modified concept attribution.

## 7 CONCLUSION

Our CoLa-DCE method generating concept-guided counterfactuals successfully tackles the lack of transparency and fine-grained control in current diffusion-based counterfactual generation methods. Starting from an improved target selection incorporating the models' perception, we show how our concept-based approach yields semantically smaller image changes qualitatively and quantitatively, enforcing the minimality requirement. With the additional level of control by selecting concepts and adding spatial constraints per concept, the counterfactual generation is more focused on small, localized feature perturbations in the image. At the same time, the image alterations are more locally confined and comprehensible due to the concept grounding. From our CoLa-DCE explanations, it is directly deducible which feature changes at which location cause the prediction change of the classifier, strongly improving the transparency and understandability to a human user. With the high degree of control in generating images with CoLa-DCE, we are confident to induce further work using fine-grained concept guidance for image alteration tasks.

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
