# OpenReview forum: "CoLa-DCE – Concept-guided Latent Diffusion Counterfactual Explanations"
_ICLR.cc/2025/Conference — Submitted to ICLR 2025_

### Official Review · Reviewer_43Sw · 2024-10-29

**Soundness:** 2
**Presentation:** 2
**Contribution:** 2
**Rating:** 3
**Confidence:** 4

**Summary:**

The paper is concerned with the problem of generating visual counterfactual explanations (VCEs) for predictive models in computer vision. Motivated by the human perception, which is based on semantic features and concepts, the authors propose to adapt an existing LDCE algorithm to incorporate concept-conditioning, resulting in Concept-guided Latent Diffusion Counterfactual Explanations (CoLa-DCE). This is achieved by utilizing Concept-wise Relevance Propagation (CRP) to extract the concepts important to the classifier and then using them as additional constraint throughout the explanation generation process. The combination of the two methods allows for additional visualizations, where the important concepts are included next to the VCE. Following previous works, the proposed approach is evaluated with 4 classifiers on the ImageNet dataset. Experiments attempt to verify the influence of the local target selection, the chosen number of concepts, spatial constraints, how the resulting explanations may help in explaining model failures and the alignment of concepts with the introduced edits.

**Strengths:**

S1. The paper succesfully incorporates concept conditioning into the LDCE pipeline without significant losses in performance.

S2. The experimental evaluation is based on diverse experiments addressing various important research questions. The main results are further supported by experiments on other datasets shown in the appendix. The shown explanations look realistic and correctly flip the model's decision to the target class.

S3. The authors clearly state the limitations of previous works and the motivation behind the introduction of CoLa-DCE. The related works section properly mentions the most significant papers from the field and the algorithms on which CoLa-DCE builds upon.

**Weaknesses:**

W1. Following the paper's narrative, the main contribution of the proposed method is its focus on `local counterfactual targets`,  `concept-guided feature changes` and `minimality by restricting the number of feature changes`. However, to the best of my understanding, the resulting explanations do not address these aspects explicitly. For example, looking at Figures 1., 6., 8. and 9., the presented counterfactuals modify the entire image and do not restrict the edits to areas indicated by the concept visualizations. I suspect that this phenomenon might be caused by a) the spatial constraints being introduced only at the semantic latent space level b) `suspending the concept conditionining for the last 50 generation steps`. Because the source code was not yet provided, I am not able to verify that by myself. Irrespectively of the cause, my biggest concern is that in the end the method still modifies the entire image and hence does not fulfill the main promise of the paper. Therefore, one cannot be certain whether the model's decision flip was actually caused by the edits inside areas connected to concepts or a combination of those and the complementary areas. I am leaning towards the latter claim, as the authors explicitly state that `suspending the conditioning towards the end of the generation, visible semantic changes are not perceivable, but the image is classified more accurately`, meaning that modifying other areas also influences the classifier. Overall, I must argue that this point requires further elaboration from the authors.

W2. Another contribution of the paper described in the first section is generally connected with the improved quality of the obtained explanations, which are `more transparent` and `more comprehensible`. I must argue that these claims are not properly supported with any of the quantitative results shown in the work. For example, Table 1. shows that CoLa-DCE is able to slightly improve FID but at a visible cost of L1, Flip Ratio and Confidence. These results suggest that the method struggles to improve upon the baseline LDCE. Figure 3.(a) shows that even when the number of concepts gets close to 200, CoLa-DCE remains inferior to LDCE, while also failing to beat it in terms of FID. Figure 3.(b) only compares the method with its variation. Further results from the Appendix (Tables 4. and 5.) suggest that the methods performance does not follow a predictable trend when increasing the number of concepts, once again struggling to beat the baseline in many cases in terms of Flip Ratio and Confidence. Some specific cases are also concerning, as one would not expect that moving from 50 to 100 concepts would drastically decrease the Flip Ratio (see Table 5.).

W3. The overall description of the method and the methods directly related to it (Section 3.2 with CRP description, Section 4 concerned with the entire pipeline) is kept at a rather high level, making it hard to understand the specific details. For example, I was not able to grasp how concept and spatial conditioning differ from each other based on the provided descriptions, i.e. from where exactly the binary constraints come from in each case. The pseudocode (Algorithm 1) combines those two into a single line and does not elaborate on the topic further. While the constraints are mentioned in line 238 (Algorithm 1), the gradient here is taken wrt. the latent representation, while Section 4.3 uses the `x` symbol, somehow indicating that constraints are performed in the input space. In addition, while the authors clearly motivate why a specific target class is chosen (`model perception`), the paper does not show whether CoLa-DCE is able to generate explanations for class pairs shown in previous works, putting in question the general applicability of the algorithm.

W4. Following W1., I would like the authors to clarify for which class the aforementioned figures show the attributions maps. Assuming that the convention from Figure 1. is followed, I do not fully understand the added value of these visualizations, as the initial attribution map for the target class is not shown in any case. Also, do the visualizations depict all concepts utilized in the explanation generation? As the quantitative results are typically shown for the number of concepts much greater than 5, I suspect that the figures mentioned in W1. only show the most important concepts, which I would argue does not fully support improving the explanation comprehensibility.

W5. Minor: the citation feature is not properly used, disabling the hyperlink functionality and forcing the reader to manually search through the referenced works.

**Questions:**

In the Weakness section, I blended the specific drawbacks of the method with the questions connected to them, which I will be happy to receive answers to. Besides a minor comment below, I would like the authors to focus on addressing the weaknesses, as they are concerned with the fundamental assumptions standing behind the introduction of the proposed algorithm. Overall, I think that the paper would benefit from more detailed descriptions of each component of the proposed approach, possibly also extending the introduced pseudocode.

C1. The authors incorrectly state that the DiME method `is limited to robust classifiers explicitly trained on noisy images`. In fact, the method is proposed for standard, non-robust classifier.

---

### Official Review · Reviewer_SphV · 2024-11-02

**Soundness:** 2
**Presentation:** 1
**Contribution:** 3
**Rating:** 3
**Confidence:** 3

**Summary:**

The paper proposes a novel system CoLa-DCE based on using latent diffusion models for visual counterfactual explanations. The main contribution of the paper is to introduce a method of concept selection and spatial conditioning to control generation process of counterfactual images. They also introduce feature visualisation tools to better identify the location of modifications. The authors demonstrate applicability of their method on multiple datasets, classifier architectures and diffusion models.

**Strengths:**

1. The proposed system is novel, further exploring use of latent diffusion models for counterfactual explanations.
2. The authors clearly illustrate applicability on multiple datasets and classifier architectures.
3. The idea to use spatial conditioning to guide the modifications is interesting.

**Weaknesses:**

1. Writing should be seriously improved.  It is really tedious to get through the paper. Often terms are not defined properly and the reader really has to rely on the context to understand used terms. This is not possible always. See a list of writing issues below:
	* I find parts of Fig. 2 unclear. The caption should describe the key features of the model. Why is there a connection between Decoder and External classifier but no arrow. What is it supposed to mean?
	* line 224 Should describe $d, \kappa$ clearly and what they denote. Is the feature encoding coming from an external network, the classifier $f$ itself etc. What is the distance metric? Euclidean distance?
	* Implicit/Explicit classifier should be described clearly when describing CoLa-DCE in Sec. 4. What is their input, output domains, what are their roles etc. I guess one of them is $f$. I assume they originate from prior literature but they also seem central to your method so need clear concise description.
	* line 249: Should define $\mathcal{N}$ or state its the set of natural numbers (can be confused with normal distribution as you use it in Eq. 1).
	* line 259 Why is the external classifier modelled as p(x|y) (which should be for diffusion model). Wouldn't the classifier be modelled as p(y|x)?
	* The notations $h, g$ almost appear out of the blue. What are their input, output domains. It is the same issue with $\delta$ but it at least has some brief description.
	* line 260. You provide absolutely no explanation of notations about concepts, how they are represented, what the binary constraints denote exactly. It is difficult to understand $\lambda_1, ..., \lambda_k$ and $\theta_1, ..., \theta_k$. Are the lambda's just subset of natural numbers from 1 to K? Your notation for $\theta$ also does not seem consistent. For starters, line 236 has it going from 0 to k while at other places it is 1 to k. More importantly, Eq. 7 makes it seem like $\theta$'s denote subset of indices but they are supposed to be binary masks. I am not sure what are these representations exactly, beyond the basic idea that they control which concepts to condition on.
	* There are two terms for datasets (line 219, 226) $X', \hat{X}$. What is the difference between the two? Is the reference data classification training/validation dataset and the other test data?
	* line 222 "As the model perception of the data shall be represented, the class predictions of the model are used to determine class affiliation." Really odd phrasing, please make it more clear.
	* line 319-320 "Using the intermediate ... high flip ratios" I am not sure how you are drawing this conclusion from Tab. 1. Please elaborate on this how you come to this conclusion.
	* line 469 What is $attr$ supposed to denote exactly. The attribution map for a particular feature/concept? If yes, why are you computing absolute magnitude for relative alignment? Is it the Frobenius norm of the difference of attributions? It should be described more clearly.
	* Please explain more clearly what the "confidence" metric is? Is it the difference between classifier's probability for the initially predicted class before and after the modifications?
        * line 295 mentions l2 norm between original and counterfactual image. Was it supposed to be a metric in Tab. 1?

2. The quantitative metrics of CoLa-DCE seem weak. LDCE seems to clearly outperform CoLA-DCE on "Flip-ratio" and "Confidence" metrics while being close on FID.

**Questions:**

Writing related questions or doubts are listed in Weaknesses. These are some other questions I have:

1. How do you obtain the influence of each concept? I couldn't find its description in Sec. 4. Please point me where it is described.

2. Did you consider comparing with [1]? I guess LDCE is your primary baseline but I would like to know your thoughts about this paper too.

3. About the claim in line 71--72 "it is often unclear which features precisely change toward the counterfactual and how they relate to the model prediction". Is this only a qualitative observation here or are there also some references to support it? While qualitatively I sort of understand this, I think it still is somewhat of an issue visually with CoLA-DCE because it visually seemed very similar to LDCE (like in Fig. 5). Some of the tools like gradient visualisation you use can still be used with LDCE (as In Fig. 5). It confuses me to what extent CoLa-DCE itself has solved this problem? Can you comment on this please. The positive for CoLa-DCE I do notice clearly is the sparsity of attribution maps.

4. Lastly, to visualise the changes did you consider simply taking the difference of the counterfactual and original images?

[1] Text-to-Image Models for Counterfactual Explanations: a Black-Box Approach. WACV 2024


Overall, despite some nice ideas and experiments, I am disappointed by the writing and not convinced by the quantitative results. Thus, I will favour rejection.

---

### Official Review · Reviewer_St7R · 2024-11-03

**Soundness:** 3
**Presentation:** 3
**Contribution:** 3
**Rating:** 5
**Confidence:** 4

**Summary:**

The author proposes a novel method, CoLa-DCE (Concept-guided Latent Diffusion Counterfactual Explanations), to generate controlled counterfactual explanations for image classification models using latent diffusion models. CoLa-DCE addresses a significant limitation of transparency in existing counterfactual generation techniques, where changes are applied and what is altered. Further, they show the usability of their method in debugging a model by making cases of misclassification more understandable.

**Strengths:**

- The paper is well written and the problem is well motivated. The current concept generation methods are lacking control, although some efforts have being made to control this generation [1, 2], but not this level of fine-grain control.
- The usability of the proposed method in diagoning model failure makes it a good contribution for evaluating robustness and fairness of practical systems.

**References: (Can be included in related work)**
1. Vendrow, Joshua, et al. "Dataset interfaces: Diagnosing model failures using controllable counterfactual generation." arXiv preprint arXiv:2302.07865 (2023).
2. Prabhu, Viraj, et al. "Lance: Stress-testing visual models by generating language-guided counterfactual images." Advances in Neural Information Processing Systems 36 (2023): 25165-25184.

**Weaknesses:**

**Major:**
- **Evaluation of the proposed method:**
  - In line 295, the authors mention using both L1 and L2 norms to evaluate the method. However, in Table 1, only the results for the L1 norm are shown.
  - While the proposed method consistently achieves the best FID score, it doesn’t surpass the baseline in most other metrics. Although the FID score is lower (indicating better quality), the difference between the best and second-best scores is relatively small. Moreover, when considering Flip Ratio (which shows how often the classifier predicts the counterfactual class), the gap between the proposed method and baseline is more noticeable.
  - Additionally, it may be helpful to consider other metrics, such as cosine similarity between CLIP embeddings (or embeddings from the target model), as these could offer better insights than the direct L1 and L2 norms between the original and generated images.

- **Lack of Human Study:** A human study could strengthen the paper, especially since the authors suggest that their method could be useful in model debugging and understanding model failure. A user-focused study demonstrating the method’s practical benefits would effectively support this claim.

**Minor:**
- The **url of the code** provided in line 100 is not working.

**Questions:**

1. Following up from the weaknesses outlined above, have you considered using additional evaluation metrics, such as cosine similarity between CLIP embeddings or targeted model embeddings, to provide a more in-depth analysis of the differences between the original and generated images?

2. Have you conducted human study to validate the usefulness of the proposed method in model debugging?

---

### Meta-Review · Area_Chair_fK2i · 2024-12-17

**Metareview:**

The paper proposes CoLa-DCE (Concept-guided Latent Diffusion Counterfactual Explanations), a novel method that generates controlled counterfactual explanations for image classification models using latent diffusion models. The method introduces concept-based spatial conditioning and incorporates Concept-wise Relevance Propagation (CRP) to determine where and which concepts to modify, addressing key limitations in existing counterfactual generation methods. CoLa-DCE is demonstrated on multiple datasets and classifiers, showcasing its utility for debugging misclassifications and improving model transparency.

Strengths

+ The method is novel, extending latent diffusion models to include concept-guided conditioning for controlled counterfactual generation.
+ CoLa-DCE demonstrates applicability across multiple datasets, classifiers, and diffusion models, indicating its broad potential.
+ The paper is well-motivated, addressing the need for transparency in counterfactual explanations and practical utility in diagnosing model failures.
+ Visual explanations produced by CoLa-DCE appear realistic and effectively flip the model’s predictions.

Weaknesses

+ Quantitative Evaluation: Table 1 only reports L1 norms despite L2 being mentioned in the text. While CoLa-DCE achieves improved FID scores, it underperforms baselines like LDCE on key metrics such as Flip Ratio and Confidence. Additional metrics (e.g., cosine similarity between embeddings) could provide better insights into the image modifications.
+ Human Study: The lack of a user study weakens claims about the method’s practical usefulness in model debugging.
+ Writing and Clarity: The paper suffers from unclear and inconsistent descriptions of key terms, methods, and notations (e.g., concept constraints, spatial conditioning, and binary masks). Figures and captions, particularly Fig. 2 and attribution maps, lack sufficient explanations. Certain claims, such as improved explanation transparency, are not fully supported by the results.
+ Key Concerns: Despite claims of minimal edits guided by concepts, the counterfactuals appear to modify entire images rather than specific areas. Suspended concept conditioning in the final generation steps raises doubts about whether changes are concept-driven. General applicability of CoLa-DCE to class pairs shown in prior works is not demonstrated.

**Additional Comments On Reviewer Discussion:**

No rebuttal and discussion.

---

### Decision · Program_Chairs · 2025-01-22

Reject